# Engineered Extracellular Vesicles with Compound-Induced Cargo Delivery to Solid Tumors

**DOI:** 10.3390/ijms24119368

**Published:** 2023-05-27

**Authors:** Raeyeong Kim, Jong Hyun Kim

**Affiliations:** Department of Biochemistry, School of Medicine, Daegu Catholic University, Daegu 42472, Republic of Korea; demisoda8989@naver.com

**Keywords:** extracellular vesicles, delivery, small molecules, protein-protein interaction, refractory cancer

## Abstract

Efficient delivery of functional factors into target cells remains challenging. Although extracellular vesicles (EVs) are considered to be potential therapeutic delivery vehicles, a variety of efficient therapeutic delivery tools are still needed for cancer cells. Herein, we demonstrated a promising method to deliver EVs to refractory cancer cells via a small molecule-induced trafficking system. We generated an inducible interaction system between the FKBP12-rapamycin-binding protein (FRB) domain and FK506 binding protein (FKBP) to deliver specific cargo to EVs. CD9, an abundant protein in EVs, was fused to the FRB domain, and the specific cargo to be delivered was linked to FKBP. Rapamycin recruited validated cargo to EVs through protein-protein interactions (PPIs), such as the FKBP-FRB interaction system. The released EVs were functionally delivered to refractory cancer cells, triple negative breast cancer cells, non-small cell lung cancer cells, and pancreatic cancer cells. Therefore, the functional delivery system driven by reversible PPIs may provide new possibilities for a therapeutic cure against refractory cancers.

## 1. Introduction

Cancer is caused by several genetic and epigenetic factors that lead to uncontrolled proliferation [1,2]. Clinically, many cancers have responded positively to conventional treatments such as surgery, chemotherapy, radiotherapy, and targeted drug therapy. Yet, these therapies have certain limitations. Chemotherapy not only causes side effects such as vomiting and myelosuppression, but also has poor bioavailability, high-dose requirements, multiple drug resistance, and non-specific targeting. Radiotherapy also leads to side effects such as radiation dermatitis and radiation pneumonitis. On the other hand, targeted drug therapy exhibits high selectivity and low cytotoxicity, but is still known to induce drug resistance [3,4,5,6]. Therefore, there is an urgent need to explore new and innovative therapeutic agents.

In general, since the bioavailability of drugs is relatively low in tumor tissue, higher doses are required, thereby increasing toxicity in normal cells and increasing the incidence of multiple drug resistance [7,8]. Therefore, a better understanding of tumor biology and the development of effective drug carrier systems such as lipids and nanoparticles that exhibit improved therapeutic effects at the tumor site are needed [9,10,11]. These carriers can either passively or actively target cancerous cells, reducing adverse side effects and improving therapeutic efficacy [12,13,14]. The emergence of nanoscale drug carriers including liposomes, polymeric nanoparticles, and micelles have demonstrated great potential clinical impact in the last two decades [15,16,17,18]. Currently, several nanoparticle-based agents are clinically approved, and many more are in various stages of clinical or preclinical development. Biocompatible lipid membranes derived from diverse adaptive cells with enhanced biocompatibility and physiochemical properties have become a popular alternative [19,20,21]. Although nanoscale drug carriers have lots of advantages as drug delivery systems, they still raise safety concerns, particularly for long-term administration due to lack of biodegradation, poor bioavailability, instability in the circulation, and potential toxicity. Proliferative or resistant cancer cells also utilize exosomes, a category of extracellular vesicles, to communicate and transfer the specific substances into neighboring cells. As the role of exosomes in cancer progression becomes more apparent, increasing effort has been put towards their clinical application in order to develop them as therapeutic agents [22].

Extracellular vesicles (EVs) are composed of lipid-bilayer-enclosed vesicles. These vesicles are released from most living cells, budding from the plasma membrane or fusing with multivesicular bodies to secrete exosomes into the extracellular environment [23,24]. EVs are essential in many physiological and pathological processes related to cancer, cardiovascular diseases, and neuronal diseases [25]. EVs contain lipids, proteins, nucleic acids, carbohydrates, and metabolites to communicate with each other and deliver functional molecules to neighboring cells. In addition, EVs play an essential role in cell communication to deliver autocrine and paracrine effects [26]. The innate abilities of EVs allow the transfer of therapeutic cargo to cells to regulate cell fate, organ phenotype, and individual health and disease. Several groups have explored the potential of EVs for delivering various therapeutic cargos, including drugs, nucleic acids, and proteins. EVs-mediated cargo delivery within pathophysiological environments such as cancers has attracted considerable attention. As such, EVs could be used as promising nanocarriers to deliver therapeutic candidates [27,28,29,30].

Protein–protein interactions (PPIs) are essential in several biological processes including cell growth, trafficking, movement, transcriptional activation, translation, and transmembrane signal transduction [31,32]. It is known that many human diseases result from abnormal PPIs, either through the loss of an essential interaction or the formation of a protein complex at an inappropriate time or location [33,34]. Therefore, modulating PPIs is a viable therapeutic strategy to cure many diseases. However, targeting PPIs for drug discovery is still challenging. Despite these challenges, several studies have provided evidence for developing small molecules that modulate PPIs, opening new opportunities in drug discovery [35,36]. In direct stabilization of PPIs, the stabilizer binds to the protein partners, which increases the mutual binding affinity. The small molecule stabilizer of a typical PPI, rapamycin, induces the formation of a complex between FK506 binding protein with a mass of 12 kDa (FKBP12) and FKBP12–rapamycin-binding (FRB) protein domain of FRAP. Rapamycin is an antifungal agent produced by *Streptomyces hygroscopicus* isolated from the Easter Island soil sample, and is known as an immunosuppressive drug, which potently inhibits the protein kinase TOR (target of rapamycin) [37,38,39]. Therefore, a two-protein interaction between FKBP12 and FRB is brought into proximity in the presence of rapamycin.

In this study, we developed a method for delivering specific cargo to EVs using compound-induced PPIs. We utilized rapamycin as a model compound that induced dimerization between FKBP12 and FRB and generated a controlled loading system wherein specific cargo was encapsulated in EVs and secreted into extracellular spaces of target cells. Finally, we produced the engineered EVs harboring functional cargo to deliver it into refractory cancer cells.

## 2. Results

### 2.1. Generation of Engineered Extracellular Vesicles with Rapamycin-Induced Delivery System

Research into using EVs as delivery vehicles for therapeutic cargo is rapidly increasing. In this study, we developed a delivery system that transfers specific cargo to EVs using a well-known PPI induced by small compounds based on comparable reports published in recent years [40,41]. We utilized biologically essential characteristics, i.e., FKBP 12 and the FRB domain of mTOR forms a heterodimer in the presence of rapamycin or its analog [42,43]. We generated a rapamycin-induced interaction system between FKBP12 and the FRB domain of mTOR. In this study, we made one construct encoding CD9, a typical EV marker, fused with EGFP-FRB and produced another construct encoding that specific cargo fused with mCherry-FKBP12 (Figure 1A). The CD9 conjugated with EGFP-FRB is recruited to the exosomal membrane and anchors the specific cargo containing mCherry-FKBP12 in the exosomes via the FKBP12 and FRB interaction in the presence of rapamycin. While CD9 connected with EGFP-FRB, protein is dissociated with the specific cargo fused with mCherry-FKBP12 in the absence of rapamycin (Figure 1B). In this study, we proposed that compound-mediated heterodimerization could induce the recruitment of specific cargo to EVs using a typical EV marker protein and PPI system.

### 2.2. CD9-EGFR-FRB and mCherry-FKBP12-Cargo Were Co-Localized in EVs by Rapamycin

In order to confirm the expression vectors and methods for EV isolation used in this study before examining the affinity of interaction between FKBP12 with FRB by rapamycin, we prepared COS-7 cells transfected with an empty vector or two plasmids encoding CD9-EGFP-FRB and mCherry-FKBP12-Cargo, respectively. The EGFP, FRB, mCherry, and FKBP12 were well expressed in COS-7 cells transfected with plasmids encoding CD9-EGFP-FRB and mCherry-FKBP12-Cargo while they were not expressed in COS-7 cells transfected with empty vector (Appendix A). Next, we isolated EVs to examine the recruitment of exogenous proteins to EVs. EV marker proteins such as Alix, CD81, CD63, TSG101, and HSP70 were detected in EVs, indicating that EV isolation methods worked well. EGFP and mCherry were also detected in the released EVs obtained from rapamycin-treated COS-7 cells (Appendix A). Next, we utilized media containing different concentrations (0, 10, 100, and 1000 nM) of rapamycin and harvested the cells at 24 h after rapamycin treatment to show the interaction affinity of FRB with FKBP12. We confirmed that all proteins were well expressed in COS-7 cells regardless of rapamycin concentration by immunoblot analysis of cell lysates. EGFP and mCherry were equally expressed, and the mCherry/GFP ratio was constant under the same conditions (Figure 2A,B). We isolated and purified the EVs to investigate the delivery efficiency to EVs under different rapamycin concentrations after the transfection of two plasmids encoding CD9-EGFP-FRB and mCherry-FKBP12-Cargo. The expression level of mCherry was slightly increased in a rapamycin dose-dependent manner, while the expression level of EGFP was constant regardless of rapamycin concentration. Several EV marker proteins including Alix, CD81, and HSP70 were stably expressed in EVs (Figure 2C,D). Next, we examined the recruitment or secretion of mCherry-FKBP12-Cargo to EVs by time kinetics of treatment at optimal concentration of rapamycin (100 nM). The expression level of mCherry increased as the treatment time increased to 24 h (Figure 2E,F). Based on these results, we found that the CD9-EGFR-FRB and mCherry-FKBP12-cargo were co-localized in EVs by rapamycin.

Next, we isolated and purified the EVs using ultracentrifugation from COS-7 cells transfected with two plasmids encoding CD9-EGFP-FRB and mCherry-FKBP12-Cargo. To visualize EVs particles, transmission electron microscopy (TEM) was used to capture EV images in the absence or presence of rapamycin. These images indicated that there was little difference between the two conditions (DMSO, or 100 nM rapamycin) with respect to the size or morphology of EVs. (Appendix A). There was a minor difference in distribution of EV particles obtained from COS-7 cells under the same conditions (Appendix A). We performed cytochemistry to examine the co-localization of CD9-EGFP-FRB and mCherry-FKBP12-Cargo by rapamycin treatment in cells. The CD9-EGFP-FRB was expressed in the cytosol and membrane fraction in the absence or presence of rapamycin. While the mCherry-FKBP12-Cargo was mainly expressed in the cytosol in the absence of rapamycin, it was expressed in the membrane fraction in the presence of rapamycin. Thus, the expression of CD9-EGFP-FRB and mCherry-FKBP12-Cargo was co-localized in the membrane in the presence of rapamycin indicating the mCherry-FKBP12-Cargo was recruited to the membrane due to an increase in the interaction of FRB and FKBP12 following rapamycin treatment (Figure 3A,B). The co-localization of CD9-EGFP-FRB and mCherry-FKBP12-Cargo was about 4-fold higher following rapamycin treatment than DMSO treatment (Figure 3C). In addition, we further investigated the co-localization of CD9-EGFP-FRB and mCherry-FKBP12-Cargo as rapamycin concentration increased. The co-localization of CD9-EGFP-FRB and mCherry-FKBP12-Cargo was increased in a rapamycin-dose-dependent manner. It was approximately 4-fold higher in 100 nM rapamycin treatment than in DMOS treatment (Figure 3D and Appendix A). Using a similar method, we checked the co-localization of CD9-EGFP-FRB and mCherry-FKBP12-Cargo as rapamycin (100 nM) treatment time was increased. It was enhanced following the 12 h treatment, showing a similar fold increase (Figure 3E and Appendix A). These results suggested that the co-localization of CD9-EGFP-FRB and mCherry-FKBP12-Cargo was increased in the membrane fraction following 100 nM rapamycin treatment, indicating that the mCherry-FKBP12-Cargo was recruited to the exosomal membrane via interaction of FRB with FKBP12 by rapamycin. Next, we switched the position of EGFP-FRB and mCherry-FKBP12 for plasmids encoding CD9 and Cargo to confirm the location effects of FRB and FKBP12 in rapamycin binding. Thus, we constructed two plasmids encoding CD9-mCherry-FKBP12 and EGFP-FRB-Cargo and performed the cytochemistry under the same conditions. The color red was shown in the membrane fraction of HEK-293 cells transfected with CD9-mCherry-FKBP12 and EGFP-FRB-Cargo while green was detected in the cytosol fraction under the same cells. The co-localization of CD9-mCherry-FKBP12 and EGFP-FRB-Cargo was increased in the membrane fraction by rapamycin treatment, indicating the EGFP-FRB-Cargo was recruited to membrane (Figure 3F). Therefore, these results suggested that rapamycin could recruit the EGFP-FRB-Cargo to a CD9-enriched membrane, such as an EV’s membrane, via FKBP12-FRB interaction.

### 2.3. The Association of CD9-EGFR-FRB with mCherry-FKBP12-Cargo Induced by Rapamycin Was Confirmed by Luciferase Assay

We utilized a luciferase assay as a reporter gene assay to validate the loading efficacy in the delivery system. We prepared a plasmid encoding mCherry-FKBP12-luciferase that replaced the specific cargo with luciferase. We performed the luciferase assay in the absence or presence of rapamycin using HEK-293 cells transfected with either mCherry-FKBP12-luciferase or CD9-EGFP-FRB and mCherry-FKBP12-luciferase. First, we measured the fluorescence signals in cells transfected with plasmids containing mCherry-FKBP12-luciferase since it is required for quantification to normalize each transfected cell. We obtained equal signals from cells expressing mCherry before performing a luciferase assay (Figure 4A). Next, we harvested the secreted EVs from the media in HEK-293 cells transfected with CD9-EGFP-FRB and mCherry-FKBP12-luciferase and then measured the luciferase activity during rapamycin treatment for 24 h. The luciferase activity from EVs was increased in a rapamycin-dose-dependent manner, indicating a 3-fold increase at 100 nM rapamycin compared to those of cells without rapamycin (Figure 4B). In addition, we measured the luciferase activity from EVs during rapamycin treatment for the different periods (6 h or 12 h). We found that the results of those cells treated with rapamycin for 12 h were similar to those results obtained for 24 h (Appendix A). When treated with 100 nM rapamycin, the luciferase activity was increased as the treatment time increased until 24 h (Figure 4C). To calculate the cargo loading to EVs via quantitative analysis, we measured the loading efficiency by dividing the luciferase activity in EVs by the luciferase activity in EVs secreted from cells. The loading efficiency in the presence of rapamycin was approximately 3-fold higher than in the absence of rapamycin (Figure 4D). This result indicated that the cargo loading is significantly increased in the presence of rapamycin, and suggested that this method might be used as a drug delivery tool via chemical induction.

### 2.4. The Rapamycin Induced the Delivery of EVs to Cancer Cells in a Dose-Dependent Manner

Ultimately, we examined whether the chemical-induced delivery system might be used as a functional delivery system that can used in the context of refractory cancer cells, such as those of non-small cell lung cancer, triple-negative breast cancer, and pancreatic cancer. First, we selected the representative refractory cancer cell line for the delivery assay. We utilized the A549, MDA-MB231, and Panc-1 as non-small cell lung cancer, triple-negative breast cancer, and pancreatic cancer cells, respectively. We treated with rapamycin at diverse concentrations (0, 10, 100, and 1000 nM) in HEK-293 cells transfected with two plasmids encoding CD9-EGFP-FRB and mCherry-FKBP12-luciferase. We isolated and purified EVs secreted from serum-free media, and then treated the EVs in refractory cancer cell lines. The signals of red fluorescence were transferred to EVs via interaction of FRB with FKBP12 by rapamycin. Their signals increased in a rapamycin-dose-dependent manner in A549, MDA-MB231, and Panc-1 cells, indicating that the EVs secreted from donor cells were delivered to refractory cancer cells (Figure 5A,B). The phase contrast images showing cell morphology of refractory cancer cells treated with EVs were unaffected in a rapamycin-dose-dependent manner (Figure 5A). We confirmed the equal number of refractory cancer cells used in this study (Figure 5C). Finally, we measured the luciferase activity from refractory cancer cells to compare the signal difference between them and red fluorescence. The luciferase activity was increased as rapamycin dosage was increased (Figure 5D). We observed a few differences between luciferase activity and red fluorescence. These results suggested that EVs containing a potential drug have the potential to be utilized as a delivery system into recipient refractory cancer cells.

## 3. Discussion

We developed a functional delivery system to efficiently transfer cargo to EVs via PPI induced by a compound. In this study, we utilized the interaction system of FKBP12 with FRB as a PPI system, rapamycin as a compound, and HEK-293 cells and COS-7 cells as donor cells, and finally refractory cancer cells such as A549, MDA-MB-231, and Panc-1 as recipient cells. As a result, these proteins and machinery coordinate the work of this system in pathophysiological conditions. The EV-based delivery system can potentially be applied in new ways to deliver biomolecules. Further developing engineered systems might improve the functional delivery capacity.

Until now, how cells incorporate proteins of interest into EVs remains largely unknown. In this study, we developed an EV-based delivery system via a PPI system induced by a compound. One report published the PPI between WW tag and Nedd4 family interacting protein 1 (Ndfip1) as delivery tools for loading to EVs [44]. The author insisted that the labeling of a target protein, Cre recombinase, with a WW tag leads to recognition by the Ndfip1, resulting in ubiquitination and loading into exosomes. The other report published a light-induced PPI system, CRY2 (photoreceptor cryptochrome 2) and CIBN (CRY-interacting basic-helix-loop helix 1 (CIB1) protein module [45]. The authors suggested that this system reversibly loads and delivers target proteins to EVs. In this study, we utilized the interaction system of FKBP12 with FRB induced by rapamycin, and a similar approach has been published previously [40,41,46]. Heath et al. established the active loading methods for encapsulating FRB-fused Cre to EVs. The active loading methods allow for the delivery of Cre recombinase to EVs via the dimerization induced by the rapalog, AP21967. However, the passive loading showed that EVs themselves could not achieve the intracellular delivery of Cre in the absence of rapalogs. Meanwhile, Somiya et al. delivered three different cargos—FRB-fused proteins, such as Cre, tTA, and LgBitT—to measure delivery activity. The delivery system depends on the fusogenic activity of VSV-G to be transferred. Thus, EVs without fusogenic proteins did not achieve successful cytoplasmic delivery of proteins of interest. These results supported that EVs generally have low fusogenic activity and cannot deliver cargo without exogenous expression of fusogenic proteins. Even though a fusogenic protein like a VSV-G improved the delivery capacity of EVs, it is an immunogenic protein due to its viral origin. Thus, VSV-G-contained EVs would be suitable in vivo, and require alternative fusogenic proteins or membrane fusion machinery to mitigate side effects. However, in this study, we established a delivery system without fusogenic proteins using a PPI system induced by rapamycin (Figure 1). In addition, the EVs isolated from HEK-293 donor cells could transfer to recipient refractory cancer cells, such as those of non-small lung cancer, triple negative breast cancer, and pancreatic cancer, demonstrating the therapeutic application of EVs (Figure 5).

In this study, we utilized two highly transfectable cell lines that can deliver products of interest. We mainly used HEK-293 cells since these cell lines are of human origin, nontoxic, nonimmunogenic in vivo [47], and cultured at high densities in large enough volumes to easily handle. Thus, the exogeneous expression of interesting proteins could transfer enormous cargo loading to EVs being used as functional carriers under appropriate conditions. These characteristics comprise an interesting potential candidate for clinical EV therapeutic applications.

Cancer includes a range of diseases with unregulated growth of malignant cells and characteristics of invasion into other body regions. Refractory cancer hardly responds to classical medical treatment, because even if it initially exhibits a positive response to treatment protocols, it will worsen quickly. Conventional therapies primarily kill cancer cells that grow and divide by interfering with DNA/RNA synthesis and inhibiting the cell cycle. In fact, the drugs are nonselective and can damage normal cells, causing unintended and undesirable side effects. In addition, since the bio-accessibility of these drugs to tumor tissues is relatively poor, higher doses are required, leading to elevated toxicity in normal cells. Therefore, it is desirable to develop therapeutic tools that can either passively or actively target tumor cells, thereby reducing adverse side effects with improved therapeutic efficacy. During the last few years, it has been reported that studies involving increased availability of versatile materials, including polymers [15,16,17,48], lipids [9,11], and polymeric hydrogels [18,49] are helpful for understanding tumor biology. These studies addressed the development of systems that can deliver chemotherapeutic agents to tumor sites with improved therapeutic efficacy. The emergence of nanotechnology for drug delivery has affected clinical therapeutics fields for several decades. Compared to conventional therapeutics, nanoscale drug carriers have demonstrated the potential to address certain challenges by improving efficacy and avoiding toxicity in normal cells. Among nanoscale drug carriers, liposome, polymeric nanoparticles, and micelles have demonstrated significant potential clinical impacts for diverse diseases [7]. At present, several nanoparticle-based therapeutics have been clinically approved, and many more are in various clinical or preclinical development stages.

## 4. Materials and Methods

### 4.1. Reagents and Antibodies

Primary antibody sources: mCherry antibody (#43590), GFP antibody (#2956), FKBP1A/FKBP12 antibody (#55104), CD9 antibody (#13174), TSG101 antibody (#72312), and Alix antibodies (#18269) were diluted 1:1000 for western blotting, which was purchased from Cell Signaling Technology (Beverly, MA, USA); mTOR (human FRB Domain) antibody (diluted 1:1000 for WB, ALX-215-065) was procured from Enzo Life Science (Farmingdale, NY, USA); HSP70 antibody (diluted 1:1000 for WB, #242707) was obtained from R&D systems (Minneapolis, MN, USA); Beta actin antibody (diluted 1:1000 for WB, ab13772) was purchased from Abcam (Cambridge, MA, USA). Secondary antibody sources: anti-rabbit (#32460) and anti-mouse secondary antibody (#32430) were obtained from Thermo Fisher Scientific (Wilmington, DE, USA).

### 4.2. DNA Plasmid

The pmCherry-CD9, pEGFP-FRB, pFKBP12-mCherry-C1 and pLenti.PGK.blast-Renilla_Luciferase were purchased from Addgene. To generate the constructs encoding CD9-EGFP-FRB, the CD9 gene was amplified by PCR using primers (Forward: 5′-GTG AAC CGT CAG ATC CGC TAG CAT GAT GCT GGT GGG CTT C and Reverse: CCT TGC TCA CCA TGG TGG CGA CCG GTG CGA CCA TCT CGC GGT TCC TGC-3′) from pmCherry-CD9 (Addgene #55013) and then cloned into pEGFP-FRB (Addgene #25919) via NheI and AgeI. For CD9-mCherry-FKBP12, the CD9 gene was amplified by PCR using primers (Forward: 5′-GAA CCG TCA GAT CCG CTA GCA TGA TGC TGG TGG GCT TCC and Reverse: TCA CCA TGG TGG CGA CCG GTC CGA CCA TCT CGC GGT TCC-3′) and inserted into pmCherry-FKBP12 (Addgene, #67900) through NheI and AgeI. Luciferase was amplified by PCR using primers (Forward: 5′-GTG AAC CGT CAG ATC C GC TAG CAT GAC TTC GAA AGT TTA TG and Reverse: CTT GCT CAC CAT GGT GGC GAC CGG TAT TTG TTC ATT TTT GAG-3′) from pLenti.PGK.blast-Renilla_Luciferase (Addgene #74444) and cloned into mCherry-pFKBP12 (Addgene plasmid 49385) through NheI and AgeI, generating mCherry-FKBP12-luciferase.

### 4.3. Cell Culture and Transfection

Human embryonic kidney-293 (HEK-293) cells (KCLB, Seoul, Republic of Korea), African green monkey kidney fibroblasts, and COS-7 cells (KCLB, Seoul, Republic of Korea), were cultured in Dulbecco’s modified Eagle’s medium (DMEM) containing 10% (*v*/*v*) fetal bovine serum (FBS, Welgene, Gyeonsan, Republic of Korea) supplemented with 1% (*v*/*v*) penicillin-streptomycin solution at 37 °C in a 5% CO_2_. A549, MDA-MD231, and Panc-1 were cultured in DMEM containing 10% FBS supplemented with 1% (*v*/*v*) penicillin-streptomycin solution. Transfection was performed: the indicated DNA plasmids were transfected into HEK-293 cells or COS-7 cells using Lipofectamine 2000 (Invitrogen, Carlsbad, CA, USA), according to the manufacturer’s instructions. At 4 h after transfection, media were changed to complete DMEM to maintain the cell lines.

### 4.4. Chemical Treatments

The COS-7 and HEK-293 cells transfected with the indicated plasmids were twice rinsed with serum-free DMEM before treatment of rapamycin. The cells were incubated with serum-free DMEM containing DMSO or rapamycin at diverse concentrations for the indicated time. Rapamycin (C_51_H_79_NO_13_, CAS No. 53123-88-9) was dissolved in DMSO ((CH_3_)_2_SO, CAS No.67-68-5). DMSO was treated as a vehicle control. DMSO and rapamycin were purchased from Sigma-Aldrich (St. Louis, MO, USA).

### 4.5. Exosome Purification and Isolation

Isolation of EVs was performed by modifying the exosome extraction method of Richard J. et al. [50]. All EV isolations were performed from a serum-free medium in transfected COS-7 and HEK293 cells. Serum-free culture media obtained from transfected cells were centrifuged using a Hanil Union 32R (Gimpo, Republic of Korea) centrifuge at 300× *g* at 4 °C for 15 min to discard detached cells. The supernatant was filtered through pore size 0.45 µM membrane filters (Sartorius Minisarts CA Syringe) to remove contaminating apoptotic bodies and cell debris. The cleaned cell culture media were centrifuged at 100,000× *g* for 120 min at 4 °C with a Type 70 Ti rotor using a Beckman Coulter OptimaTM L-90K Ultracentrifuge. The EV pellet was resuspended with 100 μL of lysis buffer.

### 4.6. Immunocytochemistry

After wiping a 15 × 15 mm cover glass with 70% ethanol, it was placed in a 24-well plate, and incubated with Poly-L-lysine solution (Sigma-Aldrich, St. Louis, MO, USA) for least 2 h. HEK-293 cells were seeded in a 24 well plate, and the cells were transfected with the expression vectors encoding the indicated plasmids using 5 μL lipofectamine 2000 as per above protocols. After 24 h, transfected cells were switched into serum-free media containing rapamycin at diverse concentrations for the indicated time. After rapamycin treatment, the cells were rinsed three times with DPBS solution and fixed with 4% paraformaldehyde for 30 min. Then, the cells were stained with DAPI (1 mg/mL, Thermo Scientific™, CAS No. 28718-90-3) diluted in DPBS solutions for 1 h. The cells were covered on slide glass with mounting solution to make sample slides. The confocal images were captured from a Nikon ECLIPSE Ti A1 confocal (Tokyo, Japan). Co-localization was analyzed by quantifying fluorescence intensity using ImageJ. Data are presented as mean ± standard deviation (SD).

### 4.7. Immunoblotting Analysis

The COS-7 cells and HEK-293 cells were harvested, washed twice with ice-cold PBS, and lysed with lysis buffer (50 mM Tris/HCl, pH 7.5, 150 mM NaCl, 1% Triton X-100, 1% Na-deoxycholate, 0.1% SDS, 2 mM EDTA, inhibitor cocktail solution such as protease and phosphatase inhibitor cocktail). The cells were incubated on ice for 30 min, and the lysates was centrifuged at 15,000× *g* for 20 min to obtain the supernatant. Isolated EVs or cell lysates were separated by 10% SDS-polyacrylamide gel electrophoresis, transferred to nitrocellulose membranes (Invitrogen, Carlsbad, USA). After blocking using 5% skimmed milk in TBST buffer (50 mM Tris/HCl, pH 7.5, 2.7 mM KCl, 137 mM, 150 mM NaCl, 0.05% Tween 20), the membranes were incubated with the indicated individual monoclonal or polyclonal antibodies. The membranes were then incubated with either anti-mouse or anti-rabbit IgG, coupled with horseradish peroxidase. Immunoreactivity was visualized via colorimetric reaction using an enhanced chemiluminescence substrate buffer (Thermo Scientific™, Waltham, MA, USA), according to the instructions of the manufacturer. The immunoblots were imaged using the Davinch-Chemi Fluoro™ Imaging System (DaVinci-K, Geumchengu, Republic of Korea). Image quantification was expressed as mean ± standard deviation (SD) by averaging three data. Images were quantified and analyzed using ImageJ.

### 4.8. Reporter Gene Assay

For the luciferase assay, HEK-293 cells and COS-7 cells were transfected for luciferase assay with expression vectors. The following protocols measured the luciferase activity from isolation of EVs and cell lysates. According to manufacturer’s instructions, the samples were resuspended with 150 μL Renilla luciferase lysis buffer (Promega). The 20 μL total sample volume was dissolved in 100 μL luciferase assay reagents containing substrate. Immediately, the sample was loaded into a white 96 well plate (SPL Life sciences). Measurements were performed using SpectraMax iD5 Microplate Reader (Molecular devices, Silicon Valley, CA, USA).

### 4.9. Fluorescence Measurement

The fluorescence of mCherry or GFP was measured in a SpectraMax iD5 Microplate Reader (Molecular devices, Silicon Valley, CA, USA). The fluorescence from mCherry was measured at 615 nm wavelength upon excitation at 575 nm wavelength. The fluorescence from GFP was measured at 525 nm wavelength upon excitation at 485 nm wavelength. The samples were loaded into a black 96 well plate (SPL Life sciences).

### 4.10. Transmission Electron Microscopy (TEM)

The EVs obtained through the EV isolation process were resuspended in 200 μL of autoclaved distilled water. All carbon-coated copper grids were negatively stained with 10 μL of 2% uranyl acetate for 5 min at RT. The EVs were dropped on a 200-mesh copper grid and dried at RT for 5 min. The grids were imaged via Hitachi 7700 transmission electron microscopy (Hitachi, Tokyo, Japan) at 50 kV. Images were captured with a side-mounted CCD camera (Gatan, Pleasanton, CA, USA).

### 4.11. Nanoparticle Tracking Analysis

Nanoparticle Tracking Analysis (NTA) from Malvern (NanoSight NS300, Worcestershire, UK) can measure EV size, concentration, and distribution in 10–1000 nm of liquid, based on Brownian motion. After thoroughly washing the syringe pump with 1X PBS, 1 mL of EVs diluted 10, 50, 100, or 500 times in PBS were added to the syringe to confirm, and an appropriate concentration was selected. The syringe was inserted into the NanoSight’s syringe pump. The EVs were injected at a flow rate at RT. Brownian motions of particles (EVs) present in the sample were subjected to a laser beam, recorded with a camera, and converted into size and concentration parameters by NTA via the Stokes-Einstein equation.

### 4.12. Statistical Analysis

Data were analyzed using one-way ANOVA following Tukey’s HSD tests. Statistical analysis was performed using GraphPad Prism software (GraphPad Inc., San Diego, CA, USA). A value of *p* < 0.05 was considered statistically significant.

## Figures and Tables

**Figure 1 ijms-24-09368-f001:**
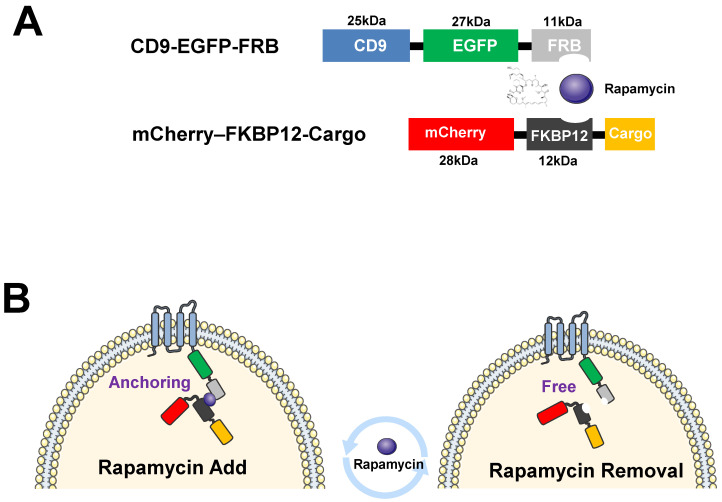
Schematic diagram illustrating genetically engineered expression vectors and extracellular vesicles (EVs) for rapamycin-induced cargo delivery. (**A**) Schematic representation of genetically engineered expression vectors of CD9-EGFP-FRB and mCherry-FKBP12-Cargo. (**B**) The schematic diagram represents the specific association between CD9-EGFP-FRB and mCherry-FKBP12-Cargo in the proposed inner vesicles with the interaction system of FRB with FKBP12 induced by rapamycin.

**Figure 2 ijms-24-09368-f002:**
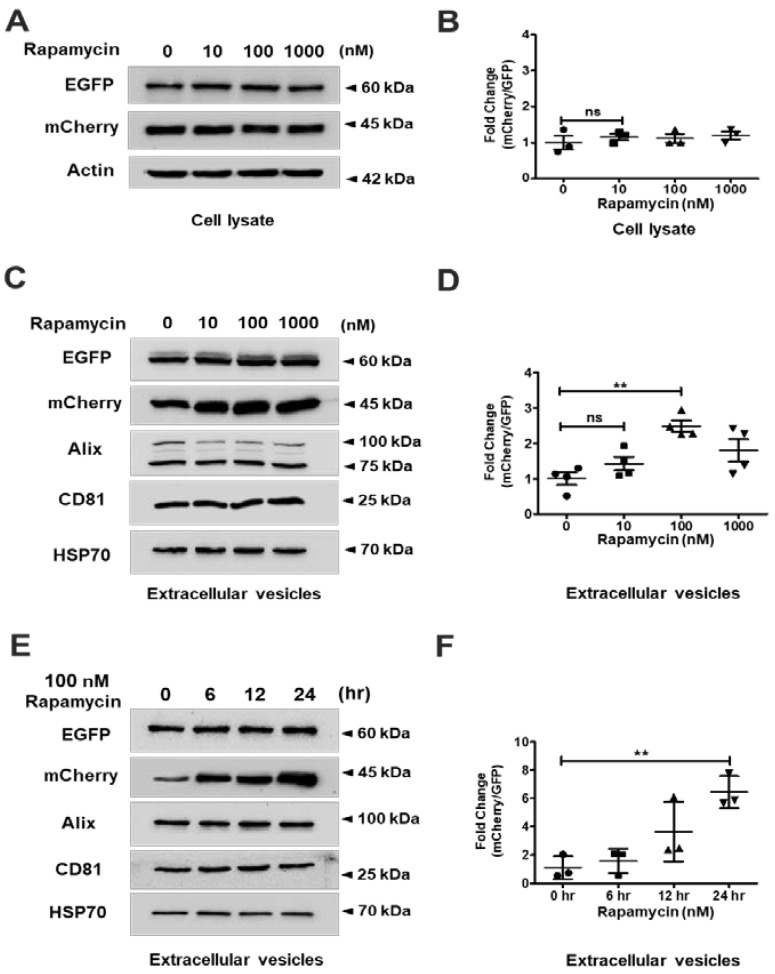
Rapamycin-induced loading of FKBP12-mCherry-Cargo into extracellular vesicles (EVs) containing CD9-EGFP-FRB. All data were analyzed via one-way ANOVA, followed by Turkey’s HSD test. (**A**) COS-7 cells were transiently transfected with the expression plasmids encoding CD9-EGFP-FRB and mCherry-FKBP12-Cargo. The cells were cultured in complete media for 24 h and then incubated with indicated rapamycin under the serum starvation condition. The expression levels of EGFP and mCherry were confirmed in cell lysates by immunoblot analysis. Actin was used as a loading control. (**B**) Quantification of panel (**A**). The quantification was performed with ImageJ. The fold change (ratio of mCherry/EGFP) was calculated between pairs of plasmid expression levels from three independent immunoblot analyses. *n* = 3, mean ± standard deviation (SD) (ns = not significant). (**C**) The EVs were isolated according to materials and methods described in the detailed isolation methods of EVs. Several proteins such as Alix, CD81, and HSP70 were enriched in EVs and used as markers. (**D**) Quantification of panel (**C**). *n* = 4, mean ± standard deviation (SD). There are significant differences between the absence or presence of rapamycin treatment groups. (ns = not significant, ** *p* = 0.0031 for rapamycin) (**E**) The transfected COS-7 cells were cultured in complete media for 24 h and then incubated with 100 nM rapamycin for different times under the serum starvation condition. The EVs were isolated according to materials and methods. (**F**) Quantification of panel (**E**). *n* = 3, mean ± standard deviation (SD). There are significant differences between 0 h and 24 h in 100 nM rapamycin treatment groups. ** *p* = 0.0047 for 24 h compared to 0 h. All raw data from immunoblots are in Appendix A.

**Figure 3 ijms-24-09368-f003:**
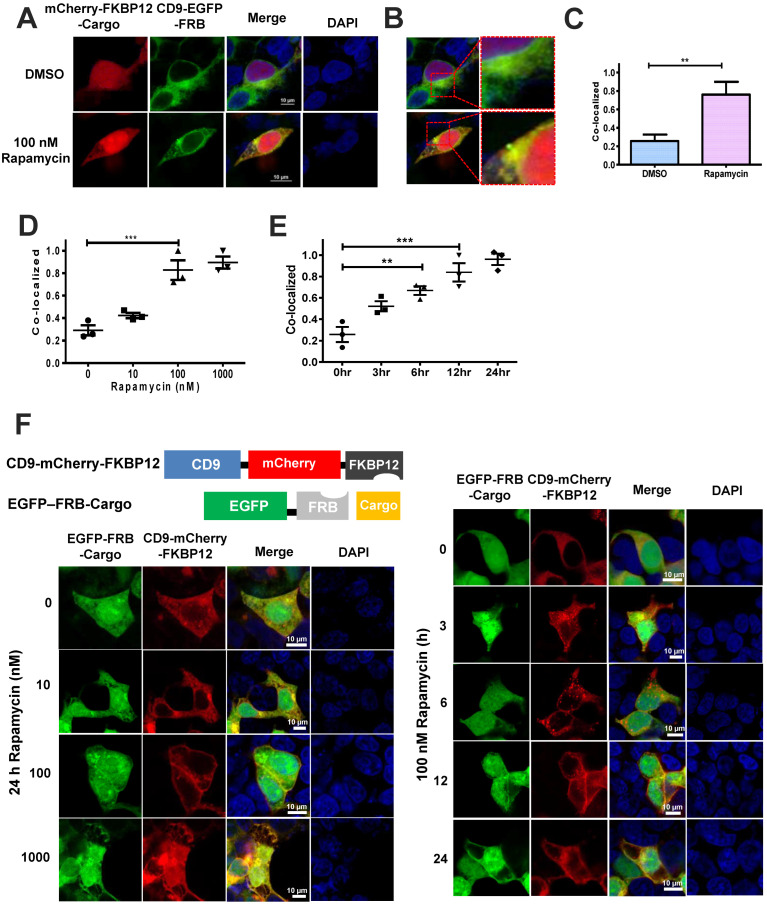
Rapamycin-induced recruitment of FKBP12-mCherry-Cargo into extracellular vesicles (EVs) expressing CD9-EGFP-FRB. The data were analyzed by one-way ANOVA, followed by Turkey’s HSD test. (**A**) HEK-293 cells were transiently transfected with the indicated plasmids. The color green indicates of CD9-EGFP-FRB and the color red indicates mCherry-FKBP12-Cargo. The merge is the co-localization of CD9-EGFP-FRB and mCherry-FKBP12-Cargo. Scale bar, 10 μm. (**B**) The magnified figure of the merged image in panel (**A**). (**C**) Quantification of the co-localization between the indicated plasmids in the absence or presence of rapamycin (100 nM). The rapamycin was dissolved in DMSO. Thus, DMSO was treated as a control while non-treated with rapamycin. ** *p* = 0.0069 for rapamycin. (**D**) The graph of panel (**C**) was plotted in a rapamycin-dose-dependent manner. The representative images are shown in Appendix A. Quantitative values were calculated from three independent experiments. There were significant differences between DMSO and 100 nM rapamycin treatment groups. *** *p* = 0.0002 for 100 nM rapamycin. (**E**) The graph in panel (**C**) was plotted while rapamycin was treated. The representative images are shown in Appendix A. There were significant differences in the absence or presence of rapamycin for 12 h. ** *p* < 0.01, *** *p* < 0.0001 treated with rapamycin for 12 h compared with that of the non-treated with rapamycin. (**F**) The schematic diagram on the left top shows the two expression vectors used in this assay. The figure in the bottom left shows the co-localization between the indicated plasmids in a rapamycin-dose-dependent manner. The figure on the right exhibits the co-localization between CD9-mCherry-FKBP12 and EGFP-FRB-Cargo at the time when treated with rapamycin. Scale bar, 10 μm.

**Figure 4 ijms-24-09368-f004:**
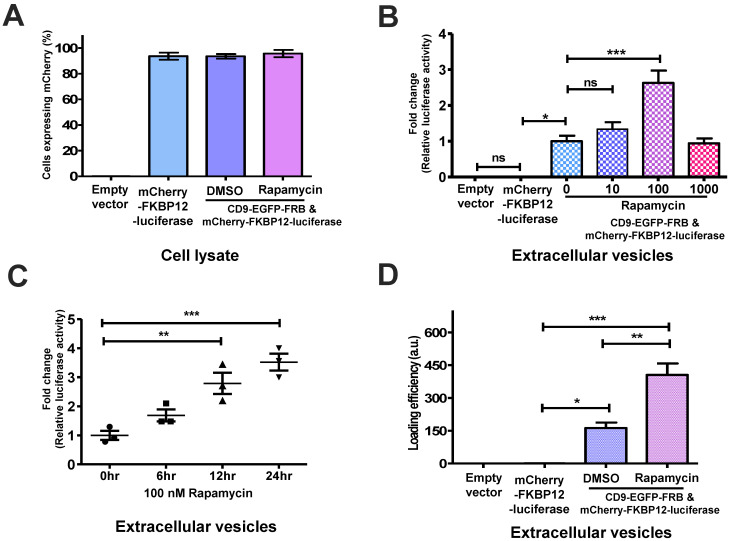
Reporter gene assay for measuring extracellular vesicle (EV)-mediated loading capacity. (**A**) The HEK-293 cells were transiently transfected with indicated plasmids. The percentage of cells expressing mCherry were counted from cell lysates after the incubation of the absence or presence of rapamycin. The data were analyzed as the mean ± standard deviation (SD) (*n* = 3). (**B**) The EVs were isolated from these transfected HEK-293 cells after incubation in the absence or presence of rapamycin. The luciferase activity was evaluated by measuring the luminescence signals of the EVs released from transfected HEK-293 cells. Mean ± standard deviation (SD) (*n* = 3). There were significant differences between their groups (ns = not significant, * *p* < 0.05, *** *p* < 0.0001). (**C**) The EVs were isolated from HEK-293 cells transfected with CD9-EGFP-FRB and mCherry-FKBP12-luciferase in the absence or the presence of rapamycin (100 nM) for the indicated time point. The data were analyzed as the mean ± standard deviation (SD) (*n* = 3) followed by Turkey’s HSD test (** *p* < 0.01, *** *p* = 0.0007). (**D**) The loading efficiency of luciferase in the EVs released from HEK-293 cells transfected with indicated plasmids. This value was calculated by dividing the number of luciferase molecules in EVs by the number of luciferase molecules in the transfected HEK-293 cells. The number of luciferase molecules was estimated from a standard curve using recombinant luciferase. The data were analyzed as the mean ± standard deviation (SD) (*n* = 3) followed by Turkey’s HSD test (* *p* < 0.05, ** *p* < 0.01, *** *p* < 0.0001).

**Figure 5 ijms-24-09368-f005:**
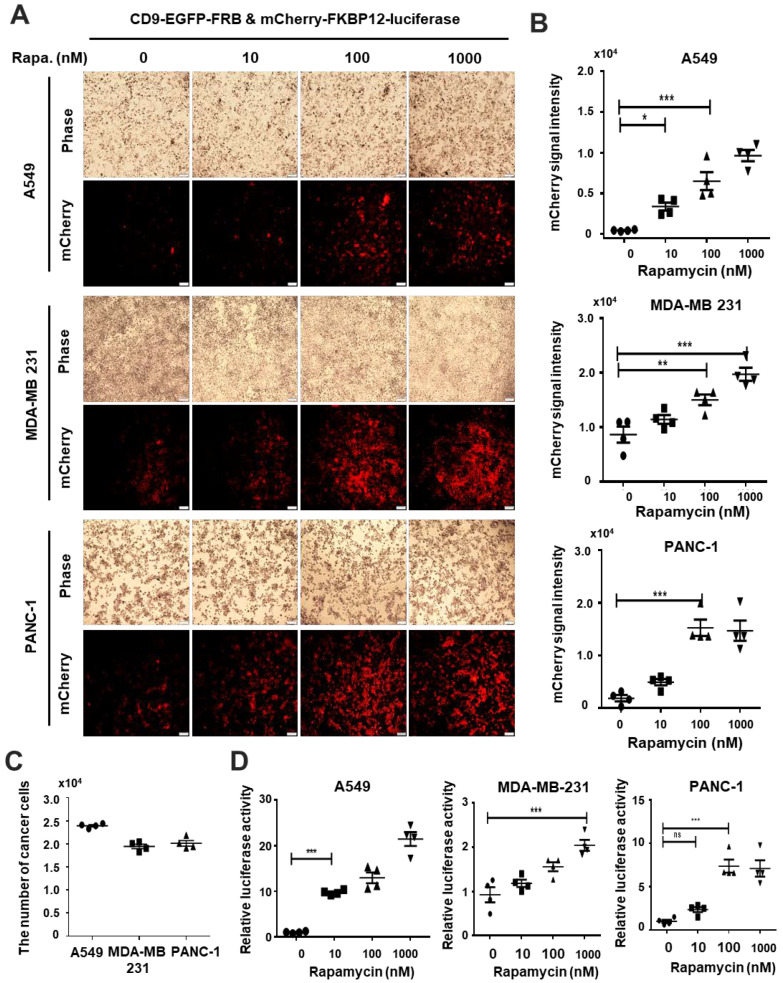
The extracellular vesicle (EV)-mediated intracellular cargo delivery into diverse cancer cell lines. (**A**) The HEK-293 cells were transfected with CD9-EGFP-FRB and mCherry-FKBP12-luciferase in the absence or presence of rapamycin. The EVs were isolated from transfected HEK-293 cells and then delivered into diverse cancer cells, such as A549, MDA-MB-231, and Panc-1 cells. After treatment, the images of phase and mCherry were captured as described above. These representative images were captured from random sites. Scale bar, 100 μm. (**B**) The signals of mCherry in panel (**A**) were quantified in a rapamycin-dose-dependent manner. The dots in panel (**B**) were plotted from four independent experiments. The data were analyzed by one-way ANOVA, followed by Turkey’s HSD test. Mean ± standard deviation (SD) (*n* = 4). There were significant differences between the groups (* *p* < 0.05, ** *p* < 0.01, *** *p* < 0.0001). (**C**) The dot graphs show the number of cancer cells used in this assay. The numbers are expressed as mean value of four independent experiments. (**D**) The dot graphs exhibit luciferase activity delivered into diverse cancer cells used as recipient cells. The luciferase activity was demonstrated by measuring the luminescent signal delivered into cancer cells. The data were analyzed by one-way ANOVA, followed by Turkey’s HSD test. Mean ± standard deviation (SD) (*n* = 4). This group had significant differences (ns = not significant, *** *p* < 0.0001).

## Data Availability

The data that support the findings of this study are available upon reasonable request from the authors.

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
