# Peer review of "Engineered Extracellular Vesicles with Compound-Induced Cargo Delivery to Solid Tumors"

_ijms, 2023, doi:10.3390/ijms24119368_

Round 1
Reviewer 1 Report
Paragraph titles often lack precision and matching with experimentally supported conclusions
Why the interaction whould occur in EVs and not in the cells?
Why the increase of mCherry in EVs is not parallelled by distribution changes at intracellular level?
How can the authors claim that fusion proteins are INTO EVs?
Controls including tratments with reporter alone are lacking.
The subcellular signal of non-treated cells is not differing from treated ones in Fugure 3A. It is not clear what has been quantified exactly in 3B and 3C.
I do not see data supporting the authors'claim about "reversible delivery".
Each cell linehas its own internalization rates, therefore authors conclusions about Figure 5 are not supported.
Not clear what Supplementary Figure 2A is.
Figure Legends are dramatically long and should meet M&Ms.
The manuscripts needs an extensive revision also in the introduction, which is out of the focus.
Author Response
Dear Dr. Editor
Thank you for giving us the opportunity to revise our manuscript entitled “Engineered extracellular vesicles with compound-induced cargo delivery to solid tumors (Manuscript Number: IJMS-2326216)”. We appreciate the reviewer’s constructive and insightful comments. We have revised the manuscript accordingly, and would like to re-submit it for your consideration. We have addressed the comments raised by the reviewers, and the amendments are highlighted in red in the revised manuscript. We hope that the revised version of the manuscript is now acceptable for publication in International Journal of Molecular Science. Point-by-point responses to the reviewers’ comments are listed below this letter.
Reviewer 1
- Paragraph titles often lack precision and matching with experimentally supported conclusions
Answer: Thank you for your kind comments. We intentionally put headings in each paragraph that were broad rather than narrow. In accordance with the reviewer’s comments, the following titles have been modified to be accurate and supportive of the results. If the revised titles of each paragraph were still not appropriate, we would appreciate the reviewer’s recommendation. We also replaced them with the following titles, and marked them in red through the entire manuscript.
2.1. Generation of engineered extracellular vesicles with rapamycin-induced delivery system
2.2 CD9-EGFR-FRB and mCherry-FKBP12-cargo were co-localized in EVs by rapamycin
2.3 Rapamycin-induced binding of mCherry-FKBP12-cargo to CD9-EGFR-FRB was confirmed by luciferase assay
2.4 The extracellular vesicles to cancer cells were delivered in a rapamycin dose-dependent manner
- Why the interaction would occur in EVs and not in the cells?
Answer: Thank you for your comments. We utilized the established the protein-protein interaction (PPI) system induced by rapamycin (Science 273: 239-242, 1996; J Am Chem Soc 127: 4715-4721, 2005). This rapamycin-induced interaction system of FRB with FKBP12 was applied to EVs delivery system in this study. This system is known as reversible interaction system since the interaction FRB with FKBP12 was dissociated in the absence of rapamycin. On the other hand, when CD9, known as tetraspanin protein with transmembrane domain, is expressed in cells, it moves to membrane, and released to extracellular matrix via EVs (Nat Commun. 11:1606, 2020; Proc Natl Acad Sci U S A. 119: e2208993119, 2022). In this study, we expressed the CD9 fusion protein linked to FRB which is able to interact with FKBP12 in response to rapamycin. In addition, CD9 fusion protein was localized in membrane and released to the extracellular matrix in the presence of rapamycin. Thus, if the CD9-EGFP-FRB and mCherry-FKBP12 were expressed in the absence of rapamycin in cells, CD9-EGFP-FRB was localized in membrane and mCherry-FKBP12 was expressed in cytol, respectively. However, if the CD9-EGFP-FRB and mCherry-FKBP12 were expressed in the presence of rapamycin, they were mainly localized in membrane fraction and then released to extracellular matrix. Therefore, we suggested that the CD9-EGFP-FRB and mCherry-FKBP12 were co-localized in membrane in EVs by the treatment of rapamycin.
- Why the increase of mCherry in EVs is not parallelled by distribution changes at intracellular level?
Answer: We did not exactly recognize which figure comments was about. However, we’ll think of it as comments on Figure 2A, and C. Figure 2A showed that the protein synthesis of mCherry was constant (at intracellular level) in the cell when rapamycin was treated. Figure 2C showed that mCherry-FKBP12 was released to EVs by the treatment of rapamycin and increased as rapamycin dosages was increased because the interaction of FKBP12 and FRB was increased in the presence of rapamycin. So, this result indicated that the increase of mCherry in EVs is not an increase of protein synthesis but rather an increase of release of EVs.
- How can the authors claim that fusion proteins are INTO EVs?
Answer: We appreciate your invaluable comments. Above mentioned, the N- or C-terminus of CD9 known as tetraspanin with transmembrane is exposed to intracellular lumen side in EVs. We made a fusion protein, C-terminus of CD9 fused with EGFP-FRB, to investigate the intracellular delivery to EVs through the protein-protein interaction induced by rapamycin. So, we described the “into EVs” to emphasize the location of the lumen side in EVs for the protein delivery. However, in order to escape the confusion to understand the meaning of “into EVs” word in manuscript, we replaced them with “to EV” word and marked them in red through the entire manuscript.
- Controls including tratments with reporter alone are lacking.
Answer: Thank for your kind comments. We confirmed that the luciferase alone as a reporter gene has no effect on cargo delivery in Figure 4 even though rapamycin was treated under the same condition. In addition, we confirmed that cargo was not delivered to EVs in transfected cells with mCherry-FKBP12-Luciferase alone as shown in Figure 4B and D.
- The subcellular signal of non-treated cells is not differing from treated ones in Figure 3A. It is not clear what has been quantified exactly in 3B and 3C.
Answer: We do not agree with you. Although the expression of CD9-EGFP-FRB was not changed in the absence or presence of rapamycin, the expression of mCherry-FKBP12-Cargo was increased in the membrane in the presence of rapamycin. So, the merged signal in membrane fraction was enhanced by the treatment of rapamycin. The yellow colors showing co-localization in merged images were analyzed via imageJ program. As shown in Figure 3C, we quantified the yellow colors in these merged images performed in at least three-independent manner. Here, we show additional images used for quantification as shown below.
- I do not see data supporting the authors'claim about "reversible delivery".
Answer: We do not agree with you. We had not described “reversible delivery” anywhere in the entire manuscript. We also do not insist on “reversible delivery” in our delivery system. The protein-protein interaction (PPI) system induced by rapamycin is known as a reversible interaction (Science 273: 239-242, 1996; J Am Chem Soc 127: 4715-4721, 2005).
- Each cell line has its own internalization rates, therefore authors conclusions about Figure 5 are not supported.
Answer: Thank you for your kind comments. We also agree with you to some context. Based on the results of this study, we isolated the rapamycin-treated EVs obtained from a dose-dependent manner and delivered to the cancer cells such as A-549, MDA-MB-231, and Panc-1, which are cancer types known to be hardly to cure. We confirmed the increase of EVs in cancer cells in a dose-dependent manner and delivery of cargo to cancer cells. Until now, even though each cancer cell has its own internalization rate, there have been few cases in which EVs have been delivered to cancer cells in rapamycin concentration-dependent manner. In this study, it was only asserted that the isolated EVs obtained from rapamycin concentration-dependent manner were delivered to various cancer cells.
- Not clear what Supplementary Figure 2A is.
Answer: Thank you for your kind comments. We isolated and purified the EVs in COS-7 cells transfected with two plasmids encoding CD9-EGFP-FRB and mCherry-FKBP12-Cargo. To visualize the EVs particles, transmission electron microscopy (TEM) was used to capture the EVs images in the absence or presence of rapamycin. These images indicated that there was little difference between the two conditions (DMSO, or 100 nM rapamycin) in the size or morphology of EVs. We clearly explained them in the results section and caption of supplemental Figure 2A. We also marked them in red through the entire manuscript.
- Figure Legends are dramatically long and should meet M&Ms.
Answer: Thank you for your suggestion. The number of words in each figure legend was efficiently reduced while avoiding confusion to understand the meaning of the sentences.
- The manuscripts needs an extensive revision also in the introduction, which is out of the focus.
Answer: We really appreciate you. In accordance with the reviewer’s comments, the discussion of cancer stem cells and refractory cancer cells was removed from the introduction as it was out of the main focus. We have inserted the new paragraphs, and references, and marked them in red through the entire manuscript.

Reviewer 2 Report
In the manuscript "Engineered extracellular vesicles with compound-induced car-2 go delivery to solid tumors" the authors describe a method to induce the dimerization of two different chimeric proteins, a protein of interest, or cargo fused to FKBP12 and CD9 linked to FRB.
CD9 is a known component of extracellular vesicle membranes; FKBP12 and FRB, instead, can interact in presence of rapamycin, thus keeping the cargo together with CD9 and shifting it in EVs. In this way, the molecule of interest can be transferred to different tumoral cell lines.
EVs are considered promising tools to deliver chemicals, proteins, RNAs and other molecules to cells in different scenarios. Protein-protein interaction (PPI) surely is a valid approach, but, as the authors themselves state, FKBP12 and FRB have been already used to load EVs with a protein cargo (Somiya, M.; Kuroda, S., Mol Pharm 2022). These researchers used CD81, which is another EV membrane protein.
the fact that Dr. Raeyeong Kim and Dr. Jong Hyun Kim achieved good cargo delivery efficiency to cancer cells in vitro alone doesn't seem enough to me to justify a new publication. The lack of novelty remains, especially considering that exosome delivery efficiency isn't improved or affected by FKBP12 and FRB.
minor points:
English should be edited throughout the manuscript.
the nature of the cargo is obscure. It should be better described.
The captions are very long, reduce them as much as possible.
figure 2
which cell line was used? HEK, as reported in the caption, or COS-7, as written in the text?
Author Response
Dear Dr. Editor
Thank you for giving us the opportunity to revise our manuscript entitled “Engineered extracellular vesicles with compound-induced cargo delivery to solid tumors (Manuscript Number: IJMS-2326216)”. We appreciate the reviewer’s constructive and insightful comments. We have revised the manuscript accordingly, and would like to re-submit it for your consideration. We have addressed the comments raised by the reviewers, and the amendments are highlighted in red in the revised manuscript. We hope that the revised version of the manuscript is now acceptable for publication in International Journal of Molecular Science. Point-by-point responses to the reviewers’ comments are listed below this letter.
Reviewer 2
In the manuscript "Engineered extracellular vesicles with compound-induced cargo delivery to solid tumors" the authors describe a method to induce the dimerization of two different chimeric proteins, a protein of interest, or cargo fused to FKBP12 and CD9 linked to FRB. CD9 is a known component of extracellular vesicle membranes; FKBP12 and FRB, instead, can interact in presence of rapamycin, thus keeping the cargo together with CD9 and shifting it in EVs. In this way, the molecule of interest can be transferred to different tumoral cell lines. EVs are considered promising tools to deliver chemicals, proteins, RNAs and other molecules to cells in different scenarios. Protein-protein interaction (PPI) surely is a valid approach, but, as the authors themselves state, FKBP12 and FRB have been already used to load EVs with a protein cargo (Somiya, M.; Kuroda, S., Mol Pharm 2022). These researchers used CD81, which is another EV membrane protein. the fact that Dr. Raeyeong Kim and Dr. Jong Hyun Kim achieved good cargo delivery efficiency to cancer cells in vitro alone doesn't seem enough to me to justify a new publication. The lack of novelty remains, especially considering that exosome delivery efficiency isn't improved or affected by FKBP12 and FRB.
Answer: Our research paper has three characteristics that distinguish it from other extracellular vesicle (EV)-related research paper. First, unlike published papers using protein-protein interaction (PPI) system with FKBP12 and FRB (Mol. Pharm. 19:2495-2505, 2022; Nanomedicine. 14:2799-2814, 2019), our research reports that EV was delivered to recipient cells without the help of other factors such as tTA, and VSG-G. Although this was slightly lower than the efficacy obtained with the help of other factors, it shows the possibility that EV delivery was possible by itself, and it was of great significance in terms of drug delivery that needs to reduce side effects. Second, CD proteins belong to the tetraspanin family, and at least 28 species were known in mammals (J. Cell Biol. 155: 1103–1108, 2001), each with unique characteristics, and their unique roles were being identified (Nat. Commun. 12:4389, 2021; J. Cell Biol. 147:599-610, 1999; J. Biol. Chem. 273:20121-20127, 1998). In particular, CD9 was highly expressed in cancer cells, and was proposed to be used as a significant prognostic marker (J. Breast Cancer. 22:77-85, 2019). Therefore, this was first research to use the PPI system for EV delivery in connection with CD9 protein, which is known to be overexpressed in diverse cancer cells, and had the advantage of applying the EV delivery systems to cancer cells in the future. Lastly, this research was the first paper to apply the EV delivery system to refractory cancers that is difficult to cure with existing treatments (see, Figure 5 in revised version). This paper suggested the possibility of treating incurable cancer with EV delivery tools beyond conventional drug treatment. In particular, it was of great significance that various cancer showed effective EV uptake. We think that we had contributed to a part of the EV research field by deriving new observation based on the previous research results.
Minor points:
- English should be edited throughout the manuscript.
Answer: Thank you for your kind comment. English language of the revised version has been extensively corrected. Changed parts in revised version manuscript were marked in red.
- the nature of the cargo is obscure. It should be better described.
Answer: We really appreciate you. Since the cargo has not yet determined in this study, any cargo can be applied. Here, we have verified that PPI system with FKBP12 and FRB is working well, and when the actual applicable cargo is determined, we will introduce and apply it to see the effect. However, we have only utilized the luciferase as a cargo in one of simple examples to measure the functionality in recipient cells.
- The captions are very long, reduce them as much as possible.
Answer: Thank you for your suggestion. The number of words in each figure legend was efficiently reduced while avoiding confusion to understand the meaning of the sentences.
- Which cell line was used in figure 2? HEK, as reported in the caption, or COS-7, as written in the text?
Answer: The mistake was corrected in figure legends. We replaced HEK-293 with COS-7 cells in figure 2 legends. Thank you for your kind comment.

Reviewer 3 Report
The manuscript proposes a new small molecule-induced trafficking system to deliver therapeutic exosomes to the refractory cancer cells.
The authors should explain better:
What chemical mechanism controls the release of exosome cargo? What therapeutic cargo was used? What is the efficiency loading and encapsulation efficiency of exosome?
Increase the size of Figure 5 B,C.D
Author Response
Dear Dr. Editor
Thank you for giving us the opportunity to revise our manuscript entitled “Engineered extracellular vesicles with compound-induced cargo delivery to solid tumors (Manuscript Number: IJMS-2326216)”. We appreciate the reviewer’s constructive and insightful comments. We have revised the manuscript accordingly, and would like to re-submit it for your consideration. We have addressed the comments raised by the reviewers, and the amendments are highlighted in red in the revised manuscript. We hope that the revised version of the manuscript is now acceptable for publication in International Journal of Molecular Science. Point-by-point responses to the reviewers’ comments are listed below this letter.
Reviewer 3
The manuscript proposes a new small molecule-induced trafficking system to deliver therapeutic exosomes to the refractory cancer cells.
The authors should explain better:
- What chemical mechanism controls the release of exosome cargo?
Answer: We utilized small compound-induced heterodimerization between the FKBP12-rapamycin-binding protein (FRB) domain and FK506 binding protein (FKBP) to sort bioactive proteins into EVs using the FRB-FKBP system. When CD9, an abundant protein in EVs, and cargo were fused with FRB and FKBP, respectively, rapamycin induced the binding of these proteins through the FKBP-FRB interaction and recruited the cargo into EVs. The released EVs delivered the cargo into recipient cells and their functionality in the recipient cells was confirmed. Thus, the FRB-FKBP system enables the delivery of functional proteins and can be used as a EVs-based protein delivery platform. Therefore, the basic principle of protein-protein interaction induced by rapamycin, a small chemical molecule, has been utilized as a mechanism for forced release of exosome cargo.
- What therapeutic cargo was used?
Answer: We really appreciate you. Since the cargo has not yet determined in this study, any cargo can be applied. Here, we have verified that PPI system with FKBP12 and FRB was working well. If the actual applicable cargo is determined, we will introduce and apply it to see the effect. However, we have only utilized the luciferase as a cargo in one of simple examples to measure the functionality in recipient cells.
- What is the efficiency loading and encapsulation efficiency of exosome?
Answer: Thank you for your kind comments. As suggested, we calculated the loading efficiency and encapsulation efficiency of exosome in this assay system. The loading efficiency and encapsulation efficiency of exosome were defined in following modified ways, respectively. We already had individual values in this assay system, but the calculations given here are approximate values of each indication.
Loading efficiency = (luciferase activity in exosome generated in the presence of rapamycin / total exosome) X 100%
Encapsulation efficiency = (luciferase activity in exosome generated in the presence of rapamycin / total luciferase activity) X 100%
As shown in Figure 4D, the luciferase activity in exosome generated in the presence of rapamycin was about 405.16 ± 74.41 and total exosome was around 2146.16 ± 121.37. Thus, the loading efficiency of exosome is around 18.88% in this assay system. The encapsulation efficiency of exosome is approximately 0.13% indicating that the luciferase activity in exosome generated in the presence of rapamycin was about 405.16 ± 74.41 and the total luciferase activity was about 296,390.88 ± 1615.89. Based on these results, the encapsulation efficiency of exosome is relatively much lower than expected data. We attribute it probably to encapsulation by natural extracellular vesicle derived from cells.
- Increase the size of Figure 5 B,C.D
Answer: Thank you for your kind comment. We replaced them with enlarged graphs showing clear numbers.

Round 2
Reviewer 1 Report
The authors have addressed several raised points and the manuscript is improved.
Reviewer 2 Report
comments in the attached file
